# Development and Evaluation of a Molecular Hepatitis A Virus Assay for Serum and Stool Specimens

**DOI:** 10.3390/v14010159

**Published:** 2022-01-15

**Authors:** Robert A. Kozak, Candace Rutherford, Melissa Richard-Greenblatt, N. Y. Elizabeth Chau, Ana Cabrera, Mia Biondi, Jamie Borlang, Jaqueline Day, Carla Osiowy, Sumathi Ramachandran, Nancy Mayer, Laurel Glaser, Marek Smieja

**Affiliations:** 1Sunnybrook Research Institute, Sunnybrook Health Sciences Centre, Toronto, ON M4N 3M5, Canada; rob.kozak@sunnybrook.ca (R.A.K.); chaun1@mcmaster.ca (N.Y.E.C.); 2St. Joseph’s Healthcare, Hamilton, ON L8N 4A6, Canada; rutherf@hhsc.ca (C.R.); Melissa.greenblatt@oahpp.ca (M.R.-G.); 3Perelman School of Medicine, University of Pennsylvania, Philadelphia, PA 19104, USA; Nancy.mayer@pennmedicine.upenn.edu (N.M.); Laurel.Glaser@pennmedicine.upenn.edu (L.G.); 4Pathology and Laboratory Medicine, London Health Sciences Centre, London, ON N6A 5W9, Canada; ana.cabrera@lhsc.on.ca; 5Toronto Centre for Liver Disease, University Health Network, Toronto, ON M6H 3M1, Canada; mia.biondi@mail.mcgill.ca; 6National Microbiology Laboratory, Winnipeg, MB R3E 3PG, Canada; jamie.borlang@canada.ca (J.B.); Jaqueline.day@canada.ca (J.D.); carla.osiowy@canada.ca (C.O.); 7Division of Viral Hepatitis, Centers for Disease Control and Prevention, Atlanta, GA 30333, USA; dcq6@cdc.gov

**Keywords:** viral hepatitis, clinical diagnostics, real-time PCR

## Abstract

Hepatitis A virus (HAV) is an emerging public health concern and there is an urgent need for ways to rapidly identify cases so that outbreaks can be managed effectively. Conventional testing for HAV relies on anti-HAV IgM seropositivity. However, studies estimate that 10–30% of patients may not be diagnosed by serology. Molecular assays that can directly detect viral nucleic acids have the potential to improve diagnosis, which is key to prevent the spread of infections. In this study, we developed a real-time PCR (RT-PCR) assay to detect HAV RNA for the identification of acute HAV infection. Primers were designed to target the conserved 5′-untranslated region (5′-UTR) of HAV, and the assay was optimized on both the Qiagen Rotor-Gene and the BD MAX. We successfully detected HAV from patient serum and stool samples with moderate differences in sensitivity and specificity depending on the platform used. Our results highlight the clinical utility of using a molecular assay to detect HAV from various specimen types that can be implemented in hospitals to assist with diagnostics, treatment and prevention.

## 1. Introduction

Hepatitis A virus is part of the genus *Hepatovirus*, within the *Picornaviridae* family. It has a single-stranded positive-sense RNA genome and is primarily transmitted by the fecal–oral route [1]. The virus causes acute infection of the liver that results in fatigue, hepatitis and jaundice. Complications can include relapsing disease and in rare cases, fulminant hepatic failure [2]. Globally the virus is a major cause of acute hepatitis [3] and is responsible for over 1.5 million cases annually. Serological studies have shown higher prevalence of hepatitis A virus (HAV) infection in many low- and middle-income countries compared with high-income countries, and many individuals are likely infected asymptomatically at a young age. However, increasing numbers of outbreaks are being seen in regions with low endemicity and minimal population immunity, due to travel and the international food trade that has resulted from globalization [4]. North America is considered a low endemicity setting with the majority of cases due to travel to high-burden countries [5], although imported cases have the potential to cause disease spread. Recent outbreaks in Canada and the USA demonstrate the importance of rapid case identification, outbreak management and contact tracing. In the 2016–18 outbreak in San Diego County, a high proportion of cases occurred among street-involved populations, most notably among people who use drugs (PWUD) and individuals experiencing homelessness [3,6]. In addition to numerous patients requiring hospitalization, a 3% mortality rate was reported among cases. In Canada, it is estimated that each symptomatic case of HAV results in CAD 1140 to 1923 of direct costs to the healthcare system [7,8]. In Ontario from 2017 to 2018, 347 cases were reported with 134 being outbreak-associated cases [9]. Of note, prior to 2018, Canadian data indicated that fewer than 300 cases were reported nationally highlighting the increase in disease burden. Moreover, recent outbreaks highlight an urgent need for rapid case identification in non-endemic countries to help prevent the spread of infections.

Infection with HAV is clinically indistinguishable from other presentations of viral hepatitis. There is only one serotype of HAV and diagnosis has traditionally relied on serological detection of IgM antibodies. However, IgM has a low predictive value in patients that do not have a hallmark clinical presentation, and the antibodies may be detected as much as six months after infection [10]. Moreover, in 10–30% of patients there is the potential for false negative and indeterminate results, especially if the individual is tested during the serological window [11], which can result in delays in case identification. This is important as timely administration of immune-globulin and vaccines can moderate disease, and limit transmission. Molecular assays have the potential to improve diagnosis and expedite public health interventions by identifying cases that are seronegative at the time of presentation [12,13].

The virus has a ~7.5 kb genome which comprises a 5′-untranslated region (5′-UTR), a single open reading frame that encodes for structural proteins (VP4, VP2, VP3, and VP1) and non-structural proteins (2A–C and 3A–D), and a 3′-untranslated region (3′-UTR) followed by a poly(A) tail(1). Based on the nucleotide sequence at the VP1/2A junction, HAV has been classified into six genotypes. Genotypes I–III are known to infect humans and there are two sub-genotypes, A and B, under each genotype [14]. The 5′-UTR is a highly conserved region, making it an ideal target for the identification of a virus that has significant genetic diversity [14,15,16]. Although laboratory-developed assays are in use, there is a paucity of molecular assays for diagnosing HAV infections that have been approved for clinical use. In this study, we developed a real-time PCR (RT-PCR) assay that detects the 5′-UTR of the HAV genome. We found that the assay can be used to detect HAV RNA from serum and stool samples, which are often easier to collect, especially from pediatric patients. This assay has the potential to be applied clinically to accurately detect multiple HAV genotypes from patient samples to improve case identification and surveillance.

## 2. Materials and Methods

### 2.1. Patient Samples

De-identified stool, plasma or serum samples from hepatitis A virus (HAV) IgM positive and negative patients were samples that were previously biobanked. Plasma samples from patients with serological evidence of hepatitis C virus (HCV) exposure or that were positive for hepatitis B virus (HBV) surface antigen were also tested as well as those from patients who were RT-PCR positive for hepatitis E virus [17]. All samples were obtained retrospectively and were included as they were ordered by the clinician for testing for HAV or for viral hepatitis (in the case of HEV, HBV and HCV positive samples), or were part of outbreak investigation (in the case of samples for the London and Philadelphia outbreaks). Specimens that were IgM negative were randomly selected within the specific time period that had sufficient volume for further testing.

### 2.2. Viruses

To investigate assay specificity, a real-time PCR (RT-PCR) was also undertaken using supernatant from cells infected with herpes simplex virus 1 (HSV1), herpes simplex virus 2 (HSV2), varicella zoster virus (VZV), cytomegalovirus (CMV), human herpesvirus 6 (HHV6), human herpesvirus 7 (HHV7) or enterovirus D68. Virus stocks of HSV1, HSV2, VZV and CMV, originally purchased as seeds from Diagnostic Hybrids, were grown in the appropriate cell lines and tested for purity using both fluorescent antibodies and lab-developed PCR before being stored at −70 °C as stocks. HS27 shell vials (Quidel, Burlington, ON, Canada) were used for varicella and cytomegalovirus, and H and V shell vials (Quidel, Burlington, ON, Canada) were used for HSV1 and 2 to prepare working concentrations of virus stocks. DNA from HHV6 and HHV7 and RNA from enterovirus D-68 was acquired from the National Microbiology Laboratory (NML) (Winnipeg, MB, Canada).

### 2.3. Bacterial Strains and Growth Conditions

*Escherichia coli* DH5α harboring the pUC57 plasmid containing the 5′-untranslated region (5′-UTR) of HAV, HA16-1496 (GenBank ID: LC373510.1) (pUC57-5′-UTR-HAV) (GenScript, Piscataway, NJ, USA) was grown in LB medium supplemented with 200 μg/mL of ampicillin and grown at 37 °C with shaking. The plasmid was purified using the Monarch Plasmid Miniprep Kit (New England BioLabs, Whitby, ON, Canada) for downstream experiments.

### 2.4. Primer Design, Nucleic Acid Extraction, and Real-Time PCR

Primers were designed using Geneious v9 [18] to target the 5′-UTR of HAV based on an alignment between 25 complete genome sequences from the National Center for Biotechnology Information (NCBI) (database queried February 2019). The alignment was manually screened for conserved regions, and the primers were screened using BLAST for non-specificity. Primer sequences selected were: HAVF2-GAGATGCCTTGGATAGGGTAAC; HAVR2-GAGACAGCCCTGACAATCAA, and primer concentrations were optimized on the Qiagen Rotor-Gene. For RT-PCR on the Qiagen Rotor-Gene, 200 μL of serum or 140 μL of a clarified stool preparation was extracted into 55 μL eluate using the NucliSENS easyMag (bioMérieux, Craponne, France). Each 12.5 μL reaction mix consisted of 6.25 μL of Quantitect SYBR Green RT-PCR master mix (Qiagen, Toronto Canada), 1.25 μL of 10× primer mix (for a final concentration of 0.2 μM per reaction) (Integrated DNA Technologies, Coralville, LA, USA), 0.125 μL Quantitect Reverse Transcriptase (Qiagen, Toronto, ON, Canada), 2.375 μL nuclease-free water (Invitrogen, Burlington, ON, Canada), and 2.5 μL of RNA template. The amplification protocol was 20 min at 50 °C, 15 min at 95 °C followed by 40 cycles of 15 s at 95 °C, 30 s at 59 °C and 20 s at 72 °C. Signal was acquired in the green (FAM) channel during extension. Amplification was followed by a melt curve running from 65 to 95 °C. The first derivative melt temperature was calculated by the instrument. For RT-PCR on the BD MAX platform, nucleic acids were extracted from 100–200 μL of serum (depending on the volume available), 1 μL of neat stool or 5 μL of 1:10 stool preparation in nuclease-free water (Invitrogen, Burlington, ON, Canada) was performed by the instrument using the BD MAX ExK TNA2 extraction kit (Beckton Dickinson, Mississauga, ON, Canada). For the clarified stool preparation, a small amount (pea-sized) of stool was diluted in 1 mL of phosphate-buffered saline (PBS), homogenized by vortexing, and then centrifuged at 10,000× *g* for 10 min. The supernatant was collected and used as template for RT-PCR. Each RT-PCR reaction consisted of 12.5 μL of Luna Universal One-Step RT-qPCR Master Mix (New England BioLabs, Canada), 2.5 μL of 10× primer mix (Integrated DNA Technologies), and 1.25 μL of reverse transcriptase (New England BioLabs, Canada). The ExK TNA2 test strip was set up according to the manufacturer’s instructions for the BD MAX. The instrument was set to workflow type 1 extraction and PCR amplification was performed for 10 min at 55 °C and 10 min at 98 °C, followed by 45 cycles of 10 s at 97 °C and 46.9 s at 60 °C. Signal was acquired in the green (FAM) channel during the annealing and extension step. Amplification was followed by a melt curve running from 60 to 95 °C and the first derivative melt temperature was calculated by the instrument.

### 2.5. Limit of Detection of RT-PCR Assay

Pooled normal human serum (Innovative Research) was spiked with pUC57-5′-UTR-HAV and a ten-fold dilution series from 10^11^ to 10^3^ copies/mL was produced. Two hundred microliters of each dilution as well as negative serum control samples were extracted in triplicate using the NucliSENS easyMag (bioMérieux, Craponne, France). The RT-PCR was performed on the Qiagen Rotor-Gene using primers HAVF1 and HAVR2 to approximate a concentration range for determining the limit of detection starting at the lowest concentration at which the plasmid was detected in 100% of the triplicate samples. Based on the RT-PCR data from the ten-fold dilution series, pUC57-5′-UTR-HAV was spiked into pooled human serum (Innovative Research, Novi, MI, USA) and a five-fold dilution series from 10^4^ to 80 copies/mL was produced. Ten replicates of 200 μL of each dilution and three replicates of negative serum samples were extracted using the NucliSENS easyMag. The RT-PCR was performed as previously described to determine the lowest concentration at which the plasmid was detected in 100% of the ten replicates.

### 2.6. Genotyping and Phylogenetic Analysis

Genotyping of the outbreak samples was performed by the Centers for Disease Control and Prevention (CDC) (Atlanta, GA, USA) or by the NML (Winnipeg, MB, Canada) in the case of banked samples by sequencing a 349 bp fragment in the VP1/2A region of the HAV genome using previously described methods [6]. Sequence analysis was undertaken with DNAStar Lasergene 11 (DNASTAR, Madison, WI, USA) and Geneious v7.0.5 (Biomatters, Aukland, New Zealand). A phylogenetic tree was constructed based on maximum likelihood algorithms [19]. Figures were generated using GraphPad Prism v6.0.

## 3. Results

### 3.1. Analytical Specificity of Primers

The HAVF1 and HAVR2 primers that were selected for the real-time PCR (RT-PCR) assay were designed based on an alignment between 25 complete hepatitis A virus (HAV) genome sequences from stool and serum isolates from different geographical regions (Table 1). Cell culture supernatant from cell lines infected with herpes simplex virus 1 (HSV1), herpes simplex virus 2 (HSV2), varicella zoster virus (VZV) or cytomegalovirus (CMV) also resulted in cross-reactivity with the HAV primers. Similarly, we also tested the HAV primers with DNA from human herpesvirus 6 (HHV6), human herpesvirus 7 (HHV7) and RNA from enterovirus D68. PCR amplification was observed with HHV7; however, the cycle threshold (Ct) value was 35.7, likely indicating a false positive.

### 3.2. Limit of Detection

To determine the limit of detection (LoD) for the detection of HAV by RT-PCR, we spiked pooled normal human serum (NHS) with known concentrations of a plasmid containing the 5′-untranslated region (5′-UTR) of HAV (pUC57-5′-UTR-HAV). We determined that the LoD of the RT-PCR assay was 4.3 × 10^2^ copies/mL.

### 3.3. Detection of HAV from Banked Patient Serum Samples

To evaluate the sensitivity and specificity of the RT-PCR assay on the Qiagen Rotor-Gene and the BD MAX using clinical specimens, we used banked serum samples from HAV IgM-positive (*n* = 77) and negative individuals (*n* = 44). Within the samples that were HAV IgM negative (*n* = 44) we included samples that were either positive for hepatitis B virus (HBV) surface antigen (*n* = 7), hepatitis C virus (HCV) antibody-positive (*n* = 3), or HEV RNA-positive (*n* = 10). We did not observe cross-reactivity with HEV RNA-positive samples.

The sensitivity of the assay was 94.8% (95% CI: 87.2–98.6) and the specificity was 100% (95% CI: 82.4–100) on the Rotor-Gene (Table 2). However, when the samples were tested on the BD MAX, the assay showed 87% (95% CI: 73.7–95.1) sensitivity and 94.7% (95% CI: 74–99.9) specificity (Table 2). In addition, the positive predictive value (PPV) and negative predictive value (NPV) were 100% and 91.7%, respectively, on the Rotor-Gene. Genotyping of HAV from these samples indicated that HAV belonging to genotypes IA, IB and IIIA were tested (Table 3). These findings demonstrate that multiple HAV genotypes can be detected from serum specimens using RT-PCR.

### 3.4. Detection of HAV from Stool Samples

In addition to testing serum samples from patients, we also wanted to determine whether our diagnostic assay can detect HAV from stool samples, as this would have significant utility in pediatric populations where sample collection is easier than venipuncture. We tested stool samples from HAV IgM-positive (*n* = 4) and HAV IgM-negative (*n* = 4) individuals and found that the sensitivity was 50% and the specificity was 100% on the Rotor-Gene. In addition, we determined that the PPV was 100% and the NPV was 66.7% (Table 2). In contrast, when the same samples were tested on the BD MAX, the sensitivity and specificity as well as the PPV and NPV were 100% (Table 2). Together, these data suggest that the sensitivity and specificity of the HAV RT-PCR assay differ depending on the clinical platform that is used when testing stool samples.

### 3.5. Clinical Evaluation of Outbreak Samples

The clinical performance of our assay was further evaluated using HAV IgM-positive (*n* = 25) and IgM-negative (*n* = 20) serum samples that were collected from the Philadelphia, Pennsylvania outbreak in 2019 (Figure 1). Testing of the samples on the Rotor-Gene and the BD MAX showed 80% sensitivity and 100% specificity on both platforms. We found that the PPV and NPV were 100% and 80%, respectively. Additionally, we also tested five samples from the 2018 outbreak in Ontario, Canada, and found that our RT-PCR assay correctly identified all the samples on both the Rotor-Gene and the BD MAX.

## 4. Discussion

Since 2017, over 30 states in the United States of America (USA) have reported hepatitis A virus (HAV) outbreaks to the Centers for Disease Control and Prevention (CDC) and case counts have totaled more than 27,000 [20]. Large outbreaks such as the one in San Diego County in 2017 highlight the importance of promptly identifying cases to provide appropriate care to patients [6,21]. Currently, the gold standard for diagnosing HAV infections relies on the presence of anti-HAV IgM antibodies. However, serological testing has low predictive value and is not always suitable for detecting recent infections [11]. Molecular assays can be used to supplement or confirm serology results, as it has been reported that viral RNA can be detected 17 days prior to and 79 days after the peak of liver enzyme levels in patients with hepatitis [19]. Here, we designed a real-time PCR (RT-PCR) assay that targets the 5′-untranslated region (5′-UTR) of HAV and is compatible with multiple platforms including the Qiagen Rotor-Gene and the BD MAX. Additionally, our assay did not cross-react with other common hepatitis viruses which could present with similar symptoms as would be seen in an acute HAV infection. Moreover, our assay is capable of detecting HAV genotypes IA, IB and IIIA from patient samples, which are the most common strains circulating in North America [3]. In 2018–19, the Canadian National Microbiology Laboratory identified the four most common genotypes in Canada. Genotypes 1a and 3a were the most prevalent (55.7% and 28.6%, respectively), but multi-jurisdictional “outbreaks” have been observed with 1a, 1b and 3a during this time. Transmission networks/outbreaks have been observed due to contaminated food, but also occur among men who have sex with men and are also associated with homelessness and injection drug use. For the Philadelphia outbreak samples, the HAV genotype was not known for all samples, which made it difficult to determine whether potential differences in the HAV genomes accounted for discrepancies in our results.

Previously, a set of RT-PCR assays were developed that used sub-type-specific primer and probe sets for detecting each HAV sub-genotype [22]. Although the assay was able to accurately identify the six different HAV sub-types from patient samples, single-plex assays can increase the turnaround time and ease of application in a clinical setting. By contrast, the California Department of Public Health developed an improved multiplex RT-PCR assay for sub-genomic identification of HAV genotypes IA, IB and IIIA, and the sensitivity and specificity were greater than 97% [23,24]. This study and others have also designed assays using FRET-based probes such as TaqMan or molecular beacon to target the VP1 capsid region or the 5′-UTR. Consistent with the design of our assay, targeting the well-conserved 5′-UTR is a common approach for molecular detection of HAV because the virus is genetically diverse [1,24]. The development of a consensus assay that can detect multiple HAV genotypes is cost-effective and allows for rapid diagnosis of infections, and a first important step will be determining an internationally agreed-upon set of primers and probes, likely targeting the 5′-UTR [25].

Interestingly, we found that the detection of HAV in serum and stool specimens on the Qiagen Rotor-Gene and BD MAX varied depending on the platform that we used. Performance differences are likely attributable to the use of different nucleic acid extraction platforms (NucliSENS easyMag vs. BD MAX), and further optimization of techniques by labs planning to carry out manual extractions may yield more consistent results. In the case of the outbreak samples, it is also possible that discrepancies in our RT-PCR data may be due to patients seroconverting but no longer being viremic. Based on our results, it is unlikely that the primers were cross-reactive with other hepatitis viruses as we did not observe amplification from serum samples that were positive for hepatitis B surface antigen or hepatitis C antibodies.

There are several limitations to our study that will be addressed in the future to further validate the RT-PCR assay. First, we had access to a limited number of stool samples for testing, and the genotypes of the viruses identified were not known. Although our data demonstrate proof of principle that the assay can be used for surveillance testing of stools (e.g., screening in outbreak settings, determining when individuals stop shedding), the paucity of samples makes it challenging to reliably determine sensitivity and specificity. Access to stool samples for HAV is relatively challenging in our region due to low prevalence, and the fact that stools are not routinely collected for diagnosis of HAV; nonetheless, additional evaluation is necessary.

Additionally, although the serum samples tested were from multiple geographical regions, we were only able to test isolates from genotypes IA, IB and IIIA. Future work will include investigating the efficacy of the RT-PCR assay for detecting other HAV genotypes and sub-genotypes. In addition, it would be informative to determine the limit of detection (LDT) of HAV in stool samples, which would complement the LDT analyses we performed in serum. Performing validation of the LDT as a quantitative assay may also be useful for monitoring viremia in patients. A study of 770 Korean patients with acute HAV infection suggested that viral load was an indicator of disease severity, as individuals with severe acute hepatitis had a >1-log higher viral load on average compared with moderate cases [26]. Further investigation to determine the specific viral load (e.g., >10^5^ copies/mL) associated with poor prognosis or greater transmission could be useful in improving clinical care and outbreak responses.

At present, detection of IgM in patient serum is the predominant method for acute case identification and has good clinical utility (https://www.who.int/news-room/fact-sheets/detail/hepatitis-a, accessed on 1 January 2022). However, serology has limitations including the potential for false negatives, possibly as a result of low antibody titers as well as the persistence of antibodies for months after an acute infection, potentially limiting the ability to perform outbreak investigation [27]. Viral RNA has been detected 14 days prior to acute serological markers, and thus molecular testing has the potential for detection of asymptomatic cases and can improve contact tracing [19]. Additionally, the potential to test stool samples may be of value for surveillance studies in low- and middle-income countries. Potentially, a hybrid approach that utilizes serology and molecular testing could be applied in patients where acute infection is suspected. In this scenario, serological screening could be performed first, and seronegative patients could be subsequently tested by RT-PCR to reduce the potential for cases being missed. Additionally, this method could help reduce false-positive cases in patients who do not present with acute hepatitis and may be an alternative to avidity testing [28]. From a clinical standpoint, this could help reduce transmission, as well as identify individuals who may benefit from post-exposure prophylaxis. However, labs will need to evaluate specimen workflow, cost and local prevalence in order to determine the feasibility of this approach.

## 5. Conclusions

In summary, we have developed a real-time PCR (RT-PCR) assay that targets the 5′-untranslated region (5′-UTR) of hepatitis A virus (HAV) and can detect multiple HAV genotypes from serum or stool specimens. The assay has been optimized for use on the Qiagen Rotor-Gene and BD MAX, which are platforms that are common to many clinical microbiology laboratories. Given the increase in frequency of HAV outbreaks in non-endemic countries, our assay shows promise for improving case identification, which can assist in an overall public health response.

## Figures and Tables

**Figure 1 viruses-14-00159-f001:**
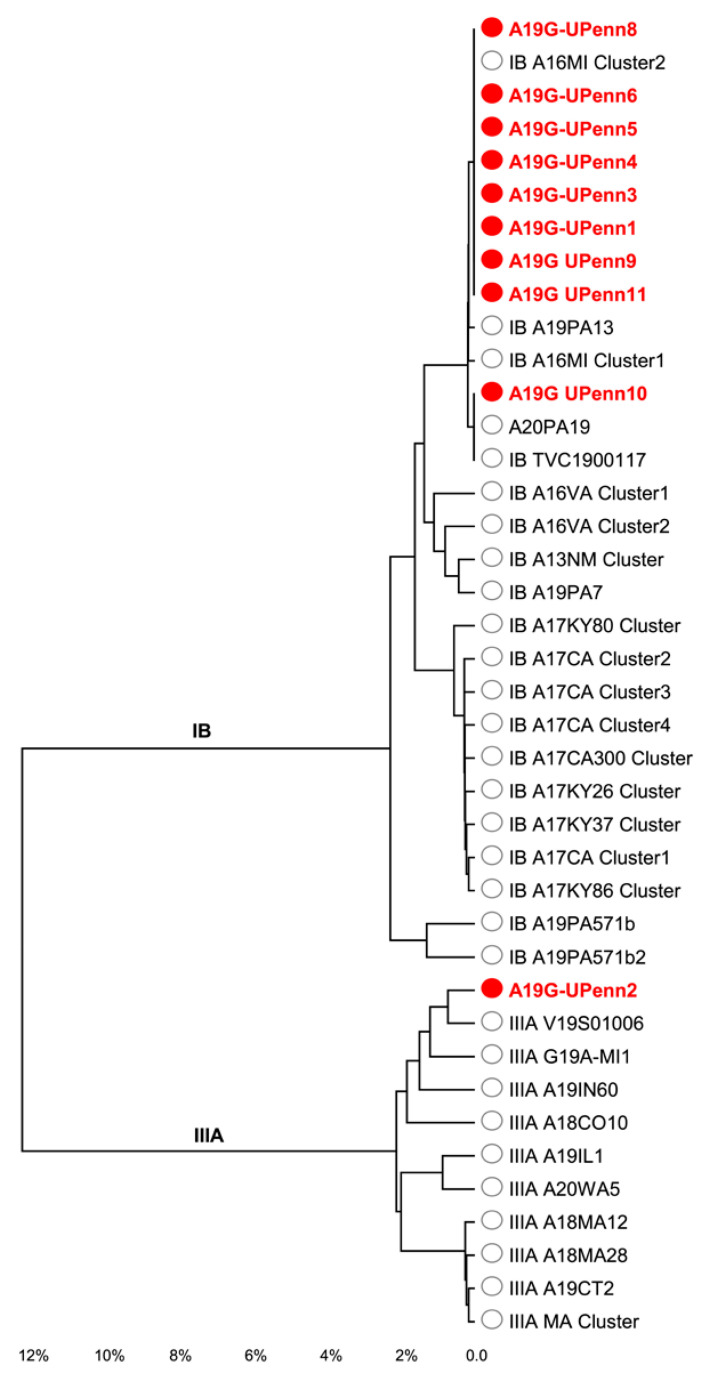
Analysis of outbreak samples. A fragment in the VP1-P2B of the HAV genome was sequenced for genotyping and a phylogenetic tree was constructed using maximum likelihood algorithms. Red circles represent sequenced isolates that were part of an outbreak identified in Philadelphia. White circles represent reference sequences.

**Table 1 viruses-14-00159-t001:** Hepatitis A genome sequences from NCBI.

GenBank ID	Source	Geographical Region
HV192266.1	Simian	Kenya
K02990.1	Human stool	USA
AB793726.1	Human serum	Japan
AB793725.1	Human serum	Japan
LC049342.1	Human serum	Mongolia
LC049339.1	Human serum	Mongolia
LC049337.1	Human serum	Mongolia
LC049341.1	Human serum	Mongolia
LC049338.1	Human serum	Mongolia
AB839695.1	Human serum	Indonesia
AB839693.1	Human serum	Indonesia
AB839697.1	Human serum	Indonesia
AB839696.1	Human serum	Indonesia
AB839694.1	Human serum	Indonesia
AB839692.1	Human serum	Indonesia
AF485328.1	Unknown	China
LC373510.1	Human serum	Japan
LC191189.1	Human serum	Japan
AB623053.1	Human serum	Japan
LC049340.1	Human serum	Mongolia
MG546668.1	Human serum	Iran
KX228694.1	Sewage	Egypt
M20273.1	Unknown	Unknown
AB258387.1	Human serum	Japan
JQ655151.1	Human stool	Korea

**Table 2 viruses-14-00159-t002:** Sensitivity and specificity of RT-PCR assay using serum and fecal samples. Confidence intervals were not calculated on groups with <10 samples.

Serum Samples (*n* = 121)	Qiagen Rotor-Gene	BD-MAX
	94.8 (95% CI: 87.2–98.6)	87 (95% CI: 73.7–95.1)
% Sensitivity		
	100.0 (95% CI: 92.0–100.0)	94.7 (95% CI: 74.0–99.9)
% Specificity		
Fecal samples (*n* = 8)	Qiagen Rotor-Gene	BD-MAX
	50 (*n* = 2/4)	100 (*n* = 4/4)
% Sensitivity		
	100 (*n* = 4/4)	100 (*n* = 4/4)
% Specificity		

**Table 3 viruses-14-00159-t003:** Hepatitis A genotype of IgM-positive serum samples.

Sample	Genotype	Sample	Genotype
1	1A	39	1B
2	1B	40	1A
3	1A	41	1A
4	1A	42	1B
5	3A	43	3A
6	1B	44	3A
7	1A	45	1B
8	3A	46	1A
9	1B	47	1A
10	1B	48	1A
11	3A	49	1A
12	3A	50	1A
13	3A	51	1A
14	1A	52	1A
15	3A	53	1A
16	3A	54	1A
17	1B	55	1A
18	1B	56	1A
19	3A	57	1A
20	1A	58	1A
21	1A	59	1A
22	1B	60	1A
23	1B	61	1A
24	1A	62	1A
25	1B	63	1A
26	3A	64	1A
27	1A	65	1A
28	1B	66	1B
29	3A	67	1B
30	1B	68	1B
31	3A	69	1B
32	1A	70	3A
33	1A	71	3A
34	3A	72	3A
35	1B	73	3A
36	1A	74	3A
37	3A	75	3A
38	1A		

## Data Availability

Not applicable.

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
