# Peer review of "Development and Evaluation of a Molecular Hepatitis A Virus Assay for Serum and Stool Specimens"

_viruses, 2022, doi:10.3390/v14010159_

Round 1
Reviewer 1 Report
This manuscript developed a real-time PCR assay targeting the 5’-UTR of HAV to diagnose acute HAV infection. The current diagnosis of hepatitis A dependent on anti-HAV IgM. However, IgM has a couple of weeks of serological window and lasts several months, thus a diagnosis tool for current infection is urgently needed. Although a lot samples were evaluated, the clinical application value of this real-time PCR assay was limited.
- Line 62 “The virus is” is redundant.
- The second paragraph of introduction is not clear in logic and organization. The application value of HAV RNA diagnosis and the significance of development of RNA detection were not clearly stated.
- Line 167 the results of cross-reactivity of the PCR assay with other pathogens were not shown. These results are important for the evaluation of detection methods.
- Line 172 the cycle threshold (ct) value of HHV7 was not shown, which was also important for assessing the performance of this assay. How did the sensitivity of 94.7% come out? 94.7% sensitivity is not good for RT-PCR assay and is also lower than the mul-tiplex RT-PCR assay developed by California Department of Public Health.
- Considering the very similar clinical symptoms of hepatitis E and Hepatitis A, the cross-reactivity with HEV should be tested.
- The Table 3 only showed the IgM of specimens. As the IgM window period and long-term of IgM positivity should be considered, the days after onset of symptoms of patients are very important and should be provided.
- How did the authors analyze the sensitivity of this method on patient serum samples? IgM remains positive for several months after the end of viral shedding, therefore IgM positivity does not represent HAV RNA positivity. Is there any other information about the 45 IgM-positive serums? And why all 45 serums were positive for HAV RNA should be explained.
- Line 208 The authors used two clinical platforms and showed discrimination in detection results in detecting fecal HAV RNA, and the two platforms had their own advantages and disadvantages in the detection of serum and feces, indicating that the detection method was not stable for application in clinical practice.
Author Response
Reviewer 1:
This manuscript developed a real-time PCR assay targeting the 5’-UTR of HAV to diagnose acute HAV infection. The current diagnosis of hepatitis A dependent on anti-HAV IgM. However, IgM has a couple of weeks of serological window and lasts several months, thus a diagnosis tool for current infection is urgently needed. Although a lot samples were evaluated, the clinical application value of this real-time PCR assay was limited.
- Line 62 “The virus is” is redundant.
We have corrected this sentence.
- The second paragraph of introduction is not clear in logic and organization. The application value of HAV RNA diagnosis and the significance of development of RNA detection were not clearly stated.
We thank the reviewer for this comment and have made significant changes to the introduction in order to make it more clear.
- Line 167 the results of cross-reactivity of the PCR assay with other pathogens were not shown. These results are important for the evaluation of detection methods.
We thank the reviewers for bringing this to our attention. Samples for HBV and HCV positive patients were included in the analysis, and we have clarified this is results section. Additionally, we have subsequently tested serum from 10 patients who are positive for hepatitis E virus by RT-PCR. Here again no cross-reactivity was detected in our assay, and we have modified the relevant sections in our manuscript to include this data.
- Line 172 the cycle threshold (ct) value of HHV7 was not shown, which was also important for assessing the performance of this assay. How did the sensitivity of 94.7% come out? 94.7% sensitivity is not good for RT-PCR assay and is also lower than the mul-tiplex RT-PCR assay developed by California Department of Public Health.
We thank the reviewer for their perspective and have added the Ct value to the manuscript. The Ct value for HHV-7 was 35.7, which is high and analysis of the primers using NCBI BLAST did not indicate any homologous sequences that could increase the likelihood of cross-reactivity. We agree on the importance of improving the sensitivity analysis and have added additional samples to the analysis. While we agree that higher sensitivity is better, we believe that our assay has potential utility, as its sensitivity is still higher than serological testing. This may offer the potential for a hybrid testing strategy employing both serological and molecular testing. We have added these comments to our discussion.
- Considering the very similar clinical symptoms of hepatitis E and Hepatitis A, the cross-reactivity with HEV should be tested.
We thank the reviewers for this excellent suggestion. We have tested serum from 10 patients who are positive for hepatitis E virus by RT-PCR. No cross-reactivity was detected in our assay, and we have modified the relevant sections in our manuscript to include this data.
- The Table 3 only showed the IgM of specimens. As the IgM window period and long-term of IgM positivity should be considered, the days after onset of symptoms of patients are very important and should be provided.
We thank the reviewer for the suggestion. However, a more detailed review of the cohort characteristics is not possible, as these are biobanked samples, many of which were collected at a national reference lab, and it will be extremely difficult to collect patient data. While we appreciate that long-term IgM positivity should be considered, all samples were collected on patients who displaying symptoms as well as risk factors for HAV exposure (as this is a requirement for testing), and the goal of our assay is to serve as a potential mechanism for rapid detection of acute cases. Additionally, it has been shown by other groups that viremia may persist for greater than 3 months (Bower et al J. Infect Dis. 2000) We agree that determining the kinetics of RNA positivity amongst HAV-infected patients using our assay would be a useful endeavor for future studies.
- How did the authors analyze the sensitivity of this method on patient serum samples? IgM remains positive for several months after the end of viral shedding, therefore IgM positivity does not represent HAV RNA positivity. Is there any other information about the 45 IgM-positive serums? And why all 45 serums were positive for HAV RNA should be explained.
We thank the reviewer for their comment. Samples were considered positive if HAV IgM was detected, as this is the currently accepted standard for case identification. All samples that were positive have genotyping done on the viral RNA extracted from the sample, which indicates they we truly positive for HAV.
- Line 208 The authors used two clinical platforms and showed discrimination in detection results in detecting fecal HAV RNA, and the two platforms had their own advantages and disadvantages in the detection of serum and feces, indicating that the detection method was not stable for application in clinical practice.
We thank the reviewer for their opinion. The differences in assay performance are likely due to differences in extraction methods, potential due to the fact that the BD Max platform was optimized for stool samples. The validation of assays on different platforms is a crucial aspect to any clinical microbiology laboratory and performing this is a routine practice. Our goal was to provide this comparative data as it may influence decision making in a clinical lab. Access to stool samples for HAV is relatively challenging in our region due to low prevalence, and the fact that stools are not routinely collected for diagnosis of HAV. Nonetheless, we have highlighted in the discussion that further work with a larger number of stool samples are required in order to have a more fulsome understanding of the assay characteristics for testing stool samples.
Reviewer 2 Report
The authors in this study have devised an rt-PCR for diagnosis of HAV infection by using primers from 5'-UTR of HAV and employed 2 plateforms for the test. rt-PCR is to detect multiple genotypes of HAV from various geographical regions of the World. I have gone over the manuscript and have following comments:
- Introduction: Authors have detailed out status of HAV in USA. However, there is no mention of changing epidemiology of HAV globally. A critical comments on evolving global epidemiology of HAV is worthwhile.
- Diagnosis of HAV: Authors have been critical of IgM anti-HAV in diagnosis of HAV. I believe IgM anti-HAV is one of the most consistent serological test for diagnosis of any of the hepatotrophic viruses and has stood its importance in diagnosis of HAV infection. Yes, there may be issues of contact tracing in subjects who are exposed and in early stages of infection and some with asymptomatic infection. The statements on the value of IgM anti-HAV in clinical diagnosis of HAV infection needs to be highlighted. The value of rt-PCR in HAV needs to be highlighted in certain situations as above.
- Authors used several viruses namely HCV, HBV and other non-hepatotrophic viruses in the system to check for cross reactivity. However, they failed to include HEV in the system and I believe it is important to exclude cross reactivity from HEV.
- Primers: no comments
- Limits of detection: no comments
- Specificity of primers: no comments.
- Serum test: the number of samples used for defining sensitivity and specificity of serum samples include 45 (positive) and 19 (negative). I believe these numbers are grossly inadequate. Also limited genotypes 1A, 1B and IIIA have been included.
- Stool samples: Here the number of stool samples tested are 3 (positive) and 4 (negative). I believe it is not possible to determine sensitivity, and specificity from these numbers.
- Clinical evaluation: Here again samples tested include 25 positive and 20 negative, with 5 others from 2018 epidemic. Numbers are exceptionally low.
Author Response
Reviewer 2:
The authors in this study have devised an rt-PCR for diagnosis of HAV infection by using primers from 5'-UTR of HAV and employed 2 plateforms for the test. rt-PCR is to detect multiple genotypes of HAV from various geographical regions of the World. I have gone over the manuscript and have following comments:
- Introduction: Authors have detailed out status of HAV in USA. However, there is no mention of changing epidemiology of HAV globally. A critical comments on evolving global epidemiology of HAV is worthwhile.
We thank the reviewer for this suggestion and have added a comment on the evolving global epidemiology in the introduction of the manuscript.
- Diagnosis of HAV: Authors have been critical of IgM anti-HAV in diagnosis of HAV. I believe IgM anti-HAV is one of the most consistent serological test for diagnosis of any of the hepatotrophic viruses and has stood its importance in diagnosis of HAV infection. Yes, there may be issues of contact tracing in subjects who are exposed and in early stages of infection and some with asymptomatic infection. The statements on the value of IgM anti-HAV in clinical diagnosis of HAV infection needs to be highlighted. The value of rt-PCR in HAV needs to be highlighted in certain situations as above.
We agree that serological testing for HAV has tremendous value in the testing of HAV, particularly in the cases of acute infection and has good clinical value. We have added this emphasis in our discussion. The reviewer is correct in important potential uses of RT-PCR and we have also highlighted this in the discussion.
- Authors used several viruses namely HCV, HBV and other non-hepatotrophic viruses in the system to check for cross reactivity. However, they failed to include HEV in the system and I believe it is important to exclude cross reactivity from HEV.
We thank the reviewer for this suggestion and have included patient serum samples that are positive for HEV RNA in our analysis and have modified the relevant sections of the manuscript.
- Primers: no comments
- Limits of detection: no comments
- Specificity of primers: no comments.
- Serum test: the number of samples used for defining sensitivity and specificity of serum samples include 45 (positive) and 19 (negative). I believe these numbers are grossly inadequate. Also limited genotypes 1A, 1B and IIIA have been included.
We thank the reviewer for their perspective. We have performed additional work and have tested an additional 30 HAV IgM positive serum samples, and 15 negative samples. These samples have been genotyped, and unfortunately include only genotypes 1A, 1B and 3A. This is due to the limited number of cases seen in Canada, and as a result a limited number of genotypes are available for testing. We agree that testing of additional genotypes would be a useful endeavour for future studies and have addressed this as a limitation in our discussion.
- Stool samples: Here the number of stool samples tested are 3 (positive) and 4 (negative). I believe it is not possible to determine sensitivity, and specificity from these numbers.
We thank the reviewer for this comment. While we were able to test an additional stool sample from an HAV IgM positive patient, we agree that a small sample size is not ideal to determine a reliable sensitivity and specificity. Our goal was to show that this can serve as important pilot data highlighting the potential of this assay for surveillance by public health (eg. children in outbreak settings) or hospital infection control (eg. determining when a worker is no longer shedding). Access to stool samples for HAV is relatively challenging in our region due to low prevalence, and the fact that stools are not routinely collected for diagnosis of HAV; nonetheless, additional evaluation is necessary. We have elaborated on this as a limitation in the discussion section of the manuscript.
- Clinical evaluation: Here again samples tested include 25 positive and 20 negative, with 5 others from 2018 epidemic. Numbers are exceptionally low.
We appreciate the perspective of the reviewer. While the number of outbreak samples are low, this is due to the relatively limited number of outbreak samples that were available to us for testing. We believe that when taken together with the data examining banked samples (shown earlier in the manuscript) that sufficient testing has been performed. Or course we plan on performing ongoing validation with any future outbreak or clinical samples that we receive.
Reviewer 3 Report
The study has important flaws. Although the introduction to the problem is well resolved, the methodology is not very precise with respect to the patients participating in the study. On the other hand, it is also poorly structured. The results are not properly presented and are difficult to interpret mainly due to a confusing methodology. The manuscript should be extensively restructured for publication.
Specific commentaries:
Line 42. A reference about major cause of acute hepatitis is needed.
Line 62. There is a typo.
Line 80. Is the last sentence of introduction a conclusion?
Line 84. Is the study prospective or retrospective? Where do the patient samples come from? What criteria were chosen for the selection of patient samples that were not IgM positive? Are they part of a cohort? Are they patients who suffered from acute hepatitis? How do the authors rule out patients with positive IgM and absence of viremia? The authors should better describe the study population since we do not know if these patients are patients with acute HAV and negative IgM serology.
On the other hand, it would not be convenient to describe the cell lines of the other pathogens in a section called “Patients samples”. It should be mentioned that these lines are to test the specificity of the PCR assay since the reader does not know why this is described.
Line 105. Two different PCR assays are mentioned in this section. Failure to state the reason why primers are tested with two different assays may confuse the reader. What is the purpose of using two assays? Perhaps it would be better to show the data from the PCR assay that had the best performance? The objective of the study is to test the performance of the primers rather than PCR platform.
Line 138. Was no probit analysis performed to establish the detection limit? This analysis allows taking into account the positive samples of the dilutions that do not have 100% positive replicates.
Line 152. What samples were genotyped? Are these outbreaks part of this study? Were only positive IgM genotyped?
Line 173. Were no replicates made to test for specificity? Making a single determination per pathogen can lead to a very marked drop in specificity. It would be necessary to repeat the test for that pathogen. The authors do not question whether it is a true cross-reaction or a random stochastic amplification. What was the exact Ct of the HHV7 positive sample?
Line 177. This paragraph is very redundant with the paragraph on line 138.
Line 193: As in specificity, the best way to obtain a sensitivity percentage would be to make replicates of the tested samples. The authors must specify how many RT-PCR positives and negatives there are in each of the populations tested (IgM positive/IgM negative). On the other hand, since the clinical characteristics of IgM negative patients are not known, it is important to know if these patients are true negatives.
Line 208: Only 7 samples tested is too little to obtain reliable sensitivity and specificity percentages and even less to draw conclusions about it.
Line 220: This part should be contained in materials and methods. It is a mess the patients who are finally part of this study. Were these patients, from a 2019 outbreak, diagnosed with HAV positive? How many of these negative IgM were positive with the primers designed in this study?
Author Response
Reviewer 3:
The study has important flaws. Although the introduction to the problem is well resolved, the methodology is not very precise with respect to the patients participating in the study. On the other hand, it is also poorly structured. The results are not properly presented and are difficult to interpret mainly due to a confusing methodology. The manuscript should be extensively restructured for publication.
Specific commentaries:
Line 42. A reference about major cause of acute hepatitis is needed.
This has been included
Line 62. There is a typo.
Corrected
Line 80. Is the last sentence of introduction a conclusion?
Corrected
Line 84. Is the study prospective or retrospective? Where do the patient samples come from? What criteria were chosen for the selection of patient samples that were not IgM positive? Are they part of a cohort? Are they patients who suffered from acute hepatitis? How do the authors rule out patients with positive IgM and absence of viremia? The authors should better describe the study population since we do not know if these patients are patients with acute HAV and negative IgM serology.
On the other hand, it would not be convenient to describe the cell lines of the other pathogens in a section called “Patients samples”. It should be mentioned that these lines are to test the specificity of the PCR assay since the reader does not know why this is described.
We thank the reviewer for these comments. All samples were obtained retrospectively and were included as they were ordered by the clinician for testing for HAV or for viral hepatitis (in the case of HEV, HBV and HCV positive samples), or were part of outbreak investigation. Specimens that were IgM negative were randomly selected from samples received in the laboratory within the specific time period that had sufficient volume for further testing. This has been included in the methods section. We have also moved the section on viral culture to separate section in the methods.
Line 105. Two different PCR assays are mentioned in this section. Failure to state the reason why primers are tested with two different assays may confuse the reader. What is the purpose of using two assays? Perhaps it would be better to show the data from the PCR assay that had the best performance? The objective of the study is to test the performance of the primers rather than PCR platform.
We appreciate the comments from the reviewer. The validation of assays on different platforms is a crucial aspect to any clinical microbiology laboratory and performing this is a routine practice. Our goal was to provide this comparative data as it may influence decision making in a clinical lab.
Line 138. Was no probit analysis performed to establish the detection limit? This analysis allows taking into account the positive samples of the dilutions that do not have 100% positive replicates.
We thank the reviewer for the suggestion and have performed a probit analysis. We have made the correction to the limit of detection in the manuscript.
Line 152. What samples were genotyped? Are these outbreaks part of this study? Were only positive IgM genotyped?
Genotyping was attempted on all clinical samples that were HAV IgM-positive. Genotyping for samples is included in table 3.
Line 173. Were no replicates made to test for specificity? Making a single determination per pathogen can lead to a very marked drop in specificity. It would be necessary to repeat the test for that pathogen. The authors do not question whether it is a true cross-reaction or a random stochastic amplification. What was the exact Ct of the HHV7 positive sample?
We appreciate the perspective of the reviewer. Analysis of the primers using NCBI BLAST did not indicate any homologous sequences that could increase the likelihood of cross-reactivity. Our evaluation of sensitivity also included samples from HCV and HBV positive patients (n=10). Additionally, we have tested our primers against serum from HEV RNA-positive patients. Collectively we believe this demonstrates the specificity of our assay.
Line 177. This paragraph is very redundant with the paragraph on line 138.
We have modified the paragraph to remove redundant information.
Line 193: As in specificity, the best way to obtain a sensitivity percentage would be to make replicates of the tested samples. The authors must specify how many RT-PCR positives and negatives there are in each of the populations tested (IgM positive/IgM negative). On the other hand, since the clinical characteristics of IgM negative patients are not known, it is important to know if these patients are true negatives.
We have included more samples for testing and it is specified in results section of the manuscript whether the banked patient samples were HAV IgM-positive or HAV IgM-negative. Additionally, the serological status of the outbreak samples was also known and is included in the results section. In the case of all patient samples the detection of HAV IgM is currently the gold standard for case identification, and thus the absence of HAV IgM would be considered a true negative by our current testing standards. We agree that there are limitations to serological testing and we have addressed this in the discussion.
Line 208: Only 7 samples tested is too little to obtain reliable sensitivity and specificity percentages and even less to draw conclusions about it.
We thank the reviewer for this comment. While we were able to test an additional stool sample from an HAV IgM positive patient, we agree that a small sample size is not ideal to determine a reliable sensitivity and specificity. Our goal was to show that this can serve as important pilot data highlighting the potential of this assay and we agree that further evaluation is needed. We have included this as a limitation in the discussion section of the manuscript.
Line 220: This part should be contained in materials and methods. It is a mess the patients who are finally part of this study. Were these patients, from a 2019 outbreak, diagnosed with HAV positive? How many of these negative IgM were positive with the primers designed in this study?
We have clarified this section, and it includes the number of IgM positive and IgM negative patient samples that were tested for both outbreaks. As the current gold-standard identifying an HAV positive case is serology, patients who were IgM positive were considered to be infected with the virus, and those that were IgM negative were not. We recognize the limitation of serological assays in identifying cases, however, we note that none of the IgM-negative patients had HAV RNA detected in their serum using our assay, which we believe demonstrates a good concordance. Nonetheless we recognize the concerns with using serology for case identification and have addressed this in the discussion.
Round 2
Reviewer 1 Report
At present, the diagnosis of hepatitis A depends on anti-HAV IgM. However, IgM has a serological window for several weeks, and technically anti-HAV IgM positivity can not indicate current hepatitis A infection. Therefore, there is an urgent need for highly sensitive HAV RNA detection as a diagnostic tool. The HAV RNA detection method established in this manuscript has high specificity, but the sensitivity needs to be improved. I think this method has limited value in clinical application. Please discuss the advantages and application value of this method in more detail.
Author Response
1) At present, the diagnosis of hepatitis A depends on anti-HAV IgM. However, IgM has a serological window for several weeks, and technically anti-HAV IgM positivity cannot indicate current hepatitis A infection. Therefore, there is an urgent need for highly sensitive HAV RNA detection as a diagnostic tool. The HAV RNA detection method established in this manuscript has high specificity, but the sensitivity needs to be improved. I think this method has limited value in clinical application. Please discuss the advantages and application value of this method in more detail.
A: We thank the reviewer for their perspective. Additionally, we have added to the final paragraph of our discussion and have addressed the potential clinical applications for molecular testing, including commenting how it might be used in conjunction with serological testing to reduce the likelihood of cases being missed (lines 302-318).
Reviewer 2 Report
I have critically reviewed the changes made in the manuscript vis-a-vis my comments. The introduction has been edited to my satisfaction. HEV-RNA samples have been included in the experiments as suggested. Further samples have been included in testing to improve the number to some extent.
Author Response
We thank the reviewer for their comments.
Reviewer 3 Report
Although the paper has improved, the authors leave some questions and suggestions unanswered. The manuscript is confusing about the number of samples tested. Table 1 shows 121 serum samples tested but another number appears in the text. The sum of all included samples is a bit confusing (45 IgM positives + 24 IgM negatives + 7 HEB positives + 3 HCV positives OR 10 RNA HEV + 25 IgM positive outbreak + 20 IgM negative outbreak). If more samples have been included, why hasn't the percentage of specificity changed? Furthermore, a patient with positive HCV serology may have spontaneously resolved the infection and be HCV RNA negative. This is meaningless in a cross-reactivity test.
The criteria used for the inclusion of patients remain unclear. One of the assays had a specificity of 94.7%, and the authors should have clarified this fact. Is it possible that this sample is IgM negative and HAV RNA positive? Hence the importance of establishing well the inclusion criteria of the study. The issue of the HHV7 positive sample has not been satisfactorily resolved either.
It would have helped the study to have included the accession numbers of the genotyped samples. Only the phylogenetic analysis of the outbreak samples is shown.
Author Response
1) Although the paper has improved, the authors leave some questions and suggestions unanswered. The manuscript is confusing about the number of samples tested. Table 1 shows 121 serum samples tested but another number appears in the text. The sum of all included samples is a bit confusing (45 IgM positives + 24 IgM negatives + 7 HEB positives + 3 HCV positives OR 10 RNA HEV + 25 IgM positive outbreak + 20 IgM negative outbreak). If more samples have been included, why hasn't the percentage of specificity changed? Furthermore, a patient with positive HCV serology may have spontaneously resolved the infection and be HCV RNA negative. This is meaningless in a cross-reactivity test.
A: We regret the confusion. Additional banked samples were tested and were used to for determine the values.in Table 2. We have modified the text to make this clearer to the reviewer (line 191-194).
2) The criteria used for the inclusion of patients remain unclear. One of the assays had a specificity of 94.7%, and the authors should have clarified this fact. Is it possible that this sample is IgM negative and HAV RNA positive? Hence the importance of establishing well the inclusion criteria of the study. The issue of the HHV7 positive sample has not been satisfactorily resolved either.
A: Selection of samples has been described in the methods and reflects either samples that were biobanked as part of routine clinical testing for suspected viral hepatitis, or as part of outbreak investigation (lines 91-97). No HAV-IgM negative samples had HAV RNA detected.
Additionally, we believe that the while the detection of HH7 by our assay is not ideal, the high Ct value likely suggests a false positive. We agree that validation should be performed by any laboratory before implementing this assay, and further investigation could be done on HHV7-positive samples to see if cross-reactivity is likely. However, considering the high sero-prevalence of HHV7 in adults, and that we did not see any false positives in our results, we believe it is unlikely that cross-reactivity with HHV7 will be a concern.
3) It would have helped the study to have included the accession numbers of the genotyped samples. Only the phylogenetic analysis of the outbreak samples is shown.
Regrettably these samples were genotyped by the public health refernce lab as part of routine public health surveillance, and accession numbers are not available.